# Research Progress of Pyroptosis in Fatty Liver Disease

**DOI:** 10.3390/ijms241713065

**Published:** 2023-08-22

**Authors:** Rongxuan Li, Weiyue Xue, Huiting Wei, Qingqing Fan, Xiang Li, Ye Qiu, Di Cui

**Affiliations:** 1Department of Physical Education, Hunan University, Changsha 410000, China; lirongxuan@hnu.edu.cn (R.L.); s212301567@hnu.edu.cn (W.X.); w1486776921@hnu.edu.cn (H.W.); fanqingqing@hnu.edu.cn (Q.F.); lixiang0824@hnu.edu.cn (X.L.); 2College of Biology, Hunan University, Changsha 410000, China; qiuye@hnu.edu.cn

**Keywords:** pyropztosis, fatty liver disease, non-alcoholic fatty liver disease, alcoholic liver disease, NLRP3, caspases, GSDMs

## Abstract

Fatty liver disease (FLD) is a clinical and pathological syndrome characterized by excessive fat deposition and even steatosis in hepatocytes. It has been proven that liver inflammation induced by fat and its derivatives are involved in the pathogenesis of FLD, while the precise mechanism still remains poorly understood. Pyroptosis is programmed inflammatory cell death driving cell swelling and membrane rupture. Pyroptosis is initiated by the activation of inflammasomes and caspases, which further cleaves and activates various gasdermins, leading to pores forming on the cell membrane and the release of pro-inflammatory factors such as interleukin (IL)-1β and IL-18. Recent studies demonstrate that pyroptosis occurs in hepatocytes, and inhibiting pyroptosis could effectively reduce fat deposition in the liver and could ameliorate inflammation from FLD, attracting our prime focus on the role of pyroptosis in FLD. In this manuscript, we reviewed the current understanding of pyroptosis in FLD development, aiming to provide new insights and potential research targets for the clinical diagnosis and intervention of FLD.

## 1. Introduction

Fatty liver disease (FLD), a condition of extra fat storage in liver, has developed globally into an urgent health threat but is often ignored or underestimated by the public, while it has been confirmed in academia as the top factor contributing to liver diseases or even hepatocellular carcinoma (HCC) [1]. The occurrence of FLD, including both non-alcoholic FLD (NAFLD) and alcoholic FLD (AFLD), has increased rapidly in recent decades and is closely related with people’s lifestyle. Moreover, FLD has posed a large financial burden to patients and the medical insurance system [2,3,4]. Although FLD does not affect liver function, nor does it cause other health issues in most cases, about 30% of patients get worse over time, ending up with cirrhosis of the liver, thereby culminating in liver failure as well as liver cancer. Heavy drinking causes AFLD, while the exact reason of NAFLD has not been understood; it is only known that energy surplus can increase its risk [5,6]. It has become a classical research theme to precisely elucidate FLD’s pathogenesis and to explore new targets for FLD clinical diagnose and treatment. Pyroptosis is programmed inflammatory cell death featuring cell swelling, membrane rupture, and the release of intracellular pro-inflammatory molecules such as IL-18 and IL-1β. Subsequently, these factors then recruit immune cells and activate immune defenses to further amplify local and systemic inflammatory reactions at the organ level [7,8]. Although pyroptosis is usually a beneficial immune response that clears pathogens from host cells, excessive pyroptosis may result in cell death, tissue damage, organ failure, and even septic shock [9]. The initial pyroptosis pathway primarily involves gasdermin D (GSDMD) as the effector protein. However, it has been firmly established that both GSDMA-C and GSDME exhibit pore-forming characteristics that mediate pyroptosis. In pyroptosis pathways, the inflammasome Nod-like receptor (NLR) Pyrin domain 3 (NLRP3) initiate the cleavage of caspases, while the downstream execution protein GSDMs serve as maneuvering effectors to form pores in the cellular membrane, facilitating the release of pro-inflammatory factors including IL-1β and IL-18. Research has confirmed that pyroptosis was involved in the pathogenesis and progression of FLD by associating the superabundant lipid deposition with the deterioration of inflammation and fibrosis in the liver [10]. Nevertheless, inhibiting pyroptosis in hepatocytes could effectively suppress liver damage induced by inflammation, making targeting pyroptosis a promising research avenue for treating FLD, but the mechanism behind this needs to be interpreted urgently and systematically [11]. In this review, we elaborated on the relevant proteins and molecular mechanisms of pyroptosis and investigated the mechanism of pyroptosis regulation in intrahepatic inflammation as well as its association with hepatocellular fat deposition, aiming to provide new insights and potential research targets for the translation of FLD clinical diagnosis and intervention.

## 2. Pyroptosis

Pyroptosis is a unique form of programmed cell death characterized by continuous cell swelling, cellular membrane rupture, and the release of cytoplasmic contents, resulting in an exacerbated and extended inflammatory response. Initially mistaken for apoptosis [12], pyroptosis was distinguished and named by Cookson and Brennan in 2001 [13,14]. Since then, significant progress has been made in understanding the molecular mechanism of pyroptosis. Studies by Shao and his team revealed that caspase-1/4/5/11 cleaves GSDMD to trigger pyroptosis, contradicting the previous belief that inflammatory caspases were solely responsible [15]. Inflammasomes, caspases, and GSDMs play crucial roles in orchestrating the pyroptosis pathways and associated cascades. These molecular interactions are effectively depicted in Figure 1.

### 2.1. Inflammasomes

The inflammasomes are multiprotein complexes or scaffolds first found in 2002 by Martinon [16]. Generally, all the inflammasomes share three main constructive components: a receptor, adaptor, and effector, and activated inflammasomes presented upon cellular infection or other stress, engaging innate immune defenses by specifically identifying and attacking pathogen-associated molecular patterns (PAMPs) or damage-associated molecular patterns (DAMPs) [17]. The receptor recognizes pathogens or other danger signals, the adaptor protein connects the receptor to the effector protein, and the effector protein activates the immune response by releasing pro-inflammatory cytokines [18]. Various inflammasomes have been identified, including the NLR family (NLRP1, NLRP2, NLRP3, NLRP6, NLRP12, NLRC4, etc.) and the PYHIN family (AIM2, IFI16, etc.), and they are named after the receptors that form the scaffold. Among these inflammasomes, the NLRP3 inflammasomes are the most extensively studied and well-understood. The scaffolding protein NLRP3 consists of three parts: the N-terminal pyrin domain (PYD); the central nucleotide binding oligomerization NAIP, CIITA, HET-E, and TP1 (NACHT) domain; and the C-terminal leucine-rich repeat (LRR), and they are responsible for mediating downstream protein interactions, self-oligomerization, and receptor recognition, respectively [19]. Like other inflammasomes, most of the receptor proteins can bind to the PYD domain of the adaptor protein’s apoptosis-associated speck-like protein that contains a CARD (ASC) through its own PYD domain to active ASC, and its own CARD domain will bind to the CARD domain of downstream caspases, directing the self-cleavage and maturation of caspases [20]. The NLRP3 inflammasomes can be activated by various stimuli, such as viruses, bacteria, free fatty acids (FFAs), reactive oxygen species (ROS), and other environmental irritants [17]. Emerging literature has confirmed that the NLRP3 inflammasomes activation occur by setting caspase-1 into action, subsequently leading to pyroptosis in multiple diseases, which implied the linkage of inflammasomes and the pathogenesis of these diseases, including FLD [21].

### 2.2. Caspases

Caspases, encoded by CASP genes, are a highly effective and evolutionary conserved family of cysteine proteases with strict substrate specificity and vitally involved in cell death and inflammation responses [22]. Typically, caspases are present in an inactive proenzyme form known as a pro-caspase and only participate in various signaling pathways when they are converted to active caspases. In mammals, 14 caspases have been identified to date, which can be divided into apoptotic caspases and inflammatory caspases based on their functions [23]. Among them, caspase-1 mediates classical pyroptosis and is activated by inflammasomes. In addition, inflammatory caspase-4/5/11 mediates the non-classical pyroptosis pathway, which is induced by lipopolysaccharide (LPS). Caspase-3 is commonly considered as the key executor of apoptosis, which can be activated by various factors such as tumor necrosis factor-α (TNF-α), chemotherapy drugs, etc. [24]. Recent studies have also shown that caspase-3 may mediate GSDME-related pyroptosis [25]. Additionally, caspase-8, an important initiator of apoptosis, was found participating in necroptosis and pyroptosis as well, thereby connecting different types of cell death [26]. The specific mechanism of pyroptosis mediated by various caspases will be discussed in detail below, and this can be observed from Figure 1.

### 2.3. GSDMs

GSDMs, a conserved protein family to make pores for pyroptosis, were initially explored in the gastrointestinal tract and dermis of mice and named gasdermins by Seaki et al. [27] In humans, there are currently six known members of the GSDM family, including GSDMA-D, GSDME (also known as DFNA5), and DFNB59, and most of which have been shown to have pore-forming activity [28]. The GSDMs involved in pyroptosis are depicted in Figure 1. With the exception of DFNB59, GSDMA~E all share two structural domains, namely the GSDM-C terminal domain and the GSDM-N terminal domain [29,30]. The full-length GSDMs could not induce pyroptosis because the interaction between the GSDM-C and GSDM-N terminal domains autoinhibits the activity of the GSDM-N terminal domain [15]. Only when the GSDM-C terminal domain is removed by proteolysis does the release of active GSDM-N terminal domain induce membrane perforation and cell content leakage [31,32,33]. GSDMD is the main executor of pyroptosis, and its activation by caspase-1/4/5/11 triggers the release of the active GSDMD-N terminal fragment. This fragment binds to phosphatidylinositol and cardiolipin on the cell membrane, tempting cell membrane perforation and serving as a channel for the extracellular release of mature IL-1β as well as IL-18 [32]. Similarly, but slightly differently, GSDME mediates pyroptosis mainly by the cleavage and activation of apoptotic caspase-3, thereby releasing the GSDME-N terminal fragment and rapidly shifting cells from apoptosis to pyroptosis [25,34]. Despite the limited research on GSDMB due to the absence of a mouse homologue, recent investigations have unveiled that GSDMB could be cleaved by granzyme A (GZMA) derived from cytotoxic lymphocytes, generating GSDMB-N terminal fragment [35]. In 2023, it was demonstrated, under the recognition and targeting of *Shigella* IpaH7.8, that GSDMB exhibited pore-forming activity and pyroptosis mechanisms [36,37]. The biological function of GSDMC remains undefined, but it has been found to mediate pyroptosis with the activation of caspase-8 [38,39]. Recent research indicates that GSDMA, the dominant GSDM in the skin, can be cleaved by streptococcal pyrogenic exotoxin B (SpeB), a protease virulence factor secreted by group A *Streptococcus* (GAS), resulting in pyroptosis [40,41]. As the pore-forming activities of GSDMs are successively reported, the functionality of DFNB59 is anticipated to be discovered in the near future.

### 2.4. Pathways of Pyroptosis

The regulation of pyroptosis involves multiple pathways, including the canonical and non-canonical pathways mediated by GSDMD, as well as the pyroptosis pathways mediated by GSDMA-C and GSDME (Figure 1). The canonical pyroptosis signaling pathway is initiated by classical inflammasomes. When PAMPs and DAMPs interact with pattern recognition receptors (PRRs), the corresponding inflammasomes, including NLRP1b, NLRP3, NLRC4, AIM2, and Pyrin, are activated [21]. NLRP3 scaffold protein then binds with ASC and caspase-1 to form the classic inflammasomes, which aggregate in the cytoplasm and subsequently directs the maturation of caspase-1. Activated caspase-1 not only activates the production of pro-inflammatory cytokines IL-1β and IL-18, but also cleaves GSDMD to generate an active GSDMD-N terminal fragment, inducing membrane perforation to form 10–15 nm non-selective pores. Ultimately, mature IL-1β and IL-18 are released, causing an extracellular inflammatory response and contributing to pyroptosis [42,43]. Non-canonical pyroptosis is primarily driven by human caspase-4/5 and mouse caspase-11. LPS from Gram-negative bacteria specifically recognizes and binds to the CARD structural domain of these caspases, leading to the oligomerization and activation of these molecules [44,45,46]. Activated caspase-4/5/11 cleaves GSDMD, releasing the GSDMD-N terminal fragment inducing cell membrane perforation and ultimately initiating pyroptosis. The fragments also trigger caspase-1 activation, and the latter cleaves pro-IL-1β and pro-IL-18, rendering them active and releasing them into the extracellular space, thereby amplifying the inflammatory response [46,47,48]. Additionally, caspase-11 can indirectly activate the NLRP3 inflammasomes and caspase-1 by cleaving pannexin-1, causing intracellular ATP to efflux and activating the P2X7 receptor, forming a K^+^ efflux channel [49,50,51]. Although caspase-4, 5, and 11 can trigger pyroptosis, they cannot facilitate pro-IL-1β and pro-IL-18 directly, unlike caspase-1, and the non-classical pyroptosis pathway still requires the participation of caspase-1 [48,52]. During *Yersinia* infection, caspase-8 elicited pyroptosis by cleaving GSDMC while diminishing apoptosis [53]. Recent studies have provided further evidence of the link between caspase-8 and GSDMC, where caspase-8 facilitates pyroptosis through the cleavage of GSDMC upon exposure to the metabolite alpha-ketoglutarate (α-KG) [39]. In hypoxic conditions, caspase-8 specifically cleaves increased GSDMC that is initiated by TNF-α, leading to pyroptosis [54]. GSDME-mediated pyroptosis is primarily initiated by caspase-3. Despite being traditionally associated with apoptosis, caspase-3 has been found by Shao Feng’s research team to selectively cleave GSDME under the influence of chemotherapy drugs [24]. The resultant GSDME-N terminal fragment triggers pyroptosis by inducing membrane perforation [25]. Furthermore, this type of pyroptosis is bound up with the expression level of GSDME; high levels of GSDME rapidly provoke pyroptosis when ignited by caspase-3, while low levels lead to apoptosis [25]. Rogers et al. also verified that caspase-3 converts apoptosis into pyroptosis by cleaving GSDME [34]. These findings illustrated that caspase-3/8 not only mediated pyroptosis, but also interconnected with apoptosis, pointing to the importance of probing the pyroptosis mechanism in the future. Additionally, granzyme B (GZMB) has been shown to cleave GSDME at the same site as caspase-3, effectuating pyroptosis, while also exciting caspase-3-mediated pyroptosis [55]. In the same year, according to Shao et al., natural killer cells and cytotoxic T lymphocytes intervened in pyroptosis through GZMA; GZMA initiated its pore-forming activity by cleaving GSDMB, and IFN-γ upregulated GSDMB protein expression in cells, further promoting pyroptosis [35]. Subsequently, following *Shigella* IpaH7.8 recognition and targeting, the pore-forming capacity and pyroptosis-inducing activity of GSDMB were once again validated [36,37]. In recent findings, it has been observed that under the induction of GAS SpeB, GSDMA released an active amino-terminal fragment capable of inserting into the membranes, thereby generating pore formation and triggering pyroptosis [40,41]. The aforementioned research has refuted the previous assumption that pyroptosis can only be mediated by caspases, unveiling that the nature of cell death is determined by substrates rather than upstream proteases. Hence, the pyroptosis pathways are diverse, with types and interconnections clearly visible in Figure 1.

**Figure 1 ijms-24-13065-f001:**
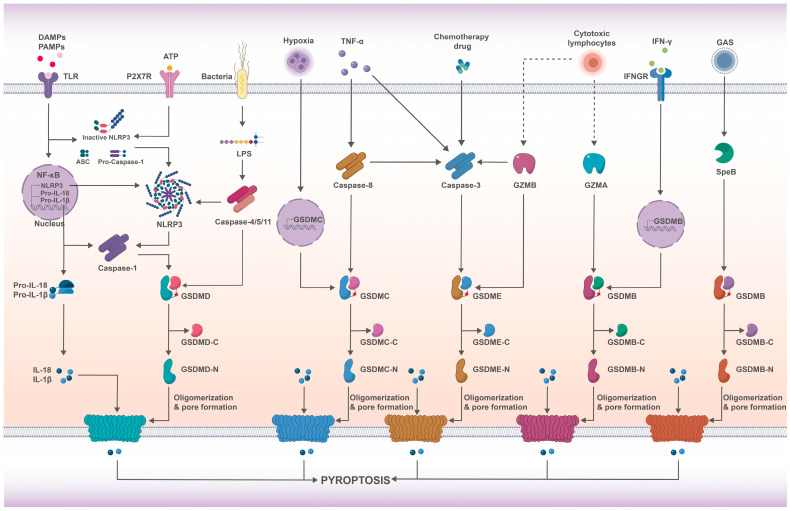
Overview of pyroptosis signaling pathways [56,57,58,59]. The classic pyroptosis pathway is initiated by DAMPs, PAMPs, and ATP, which activate TLR and P2X7R, leading to the assembly and activation of inflammasomes. Inflammasome activation induces caspase-1, which cleaves GSDMD to generate an active N-terminal fragment, causing membrane perforation and ultimately leading to the discharge of cytosolic contents and cytokines IL-18 and IL-1β. The non-classical pyroptosis pathway is induced by LPS and activates caspase-4/5/11. Similar to caspase-1, these caspases can cleave GSDMD to trigger membrane perforation and pyroptosis. Caspase-3, which is activated by caspase-8, TNF-α, chemical drugs, and GZMB, can cleave GSDME to form GSDME-N terminal fragment, leading to membrane perforation and pyroptosis. GZMB can directly cleave GSDME, inducing pyroptosis. In addition, GSDMC is responsible for the execution of caspase-8-induced pyroptosis. Under the activation of GZMA, the active GSDMB-N terminal fragment executes pyroptosis. GAS secreted a protease virulence factor, SpeB, to cleave GSDMA and induce cellular pyroptosis. DAMPs: damage-associated molecular patterns. PAMPs: pathogen-associated molecular patterns. TLR: Toll-Like Receptor. ATP: adenosine triphosphate. P2X7R: P2X7 receptor. NF-κB: nuclear factor-kappa B. LPS: lipopolysaccharide. GSDMB/C/D/E: gasdermin B/C/D/E. IL: interleukin. LPS: lipopolysaccharides. NLRP3: NLR family pyrin domain-containing 3. IFN-γ: Interferon-γ. IFNGR: IFN-g receptor. GZMA: granzyme A. GZMB: granzyme B. ASC: apoptosis-associated speck-like protein containing a CARD. TNF-α: Tumer necrosis factor-α.

## 3. Pyroptosis and FLD

FLD, a chronic liver condition with complex treatment requirements, is defined by the accumulation of fat and the resulting chronic inflammation in the liver, which underlies its fundamental pathogenesis. Pyroptosis, as a distinctive form of programmed inflammatory cell death, is likely to exert a crucial role in the intricate pathogenesis of FLD. Mounting evidence supports the notion that pyroptosis constitutes a significant pathological factor in both NAFLD and alcoholic liver disease (ALD) [11,60,61], as summarized in Table 1. After hepatocytes undergo pyroptosis, the cell membranes dissolve and rupture, releasing a large amount of inflammatory mediators that accelerate the process of liver inflammation and fibrosis [62,63]. Therefore, the role of pyroptosis and its essential components, including inflammasomes, caspases, and GSDMs, is considered crucial in initiating and advancing the pathogenesis of NAFLD and ALD. This relationship is clearly depicted in Figure 2.

### 3.1. Inflammasomes and FLD

Inflammasomes facilitating caspase-1 activation and key pro-inflammatory cytokines maturation have been implicated in the development of FLD and other related liver disease. In NAFLD, inflammasomes, particularly NLRP3, exert a substantial impact on the progression of the liver disease, ranging from steatosis to nonalcoholic steatohepatitis (NASH) and subsequently to liver fibrosis, by orchestrating pyroptosis. Although research on other inflammasomes such as NLRP6 and NLRC4 in NAFLD is limited, their role in the pathogenesis and progression of NAFLD cannot be ignored. Studies have demonstrated that *Nlrp3*^−/−^, *Asc*^−/−^, and *Casp-1*^−/−^ mice fed with a high-fat diet (HFD) exhibited diminished weight gain, reduced hepatic steatosis development (particularly in *Asc*^−/−^ mice), improved quality of epididymal adipose tissue with decreased adipocyte size, and enhanced insulin sensitivity [64]. These findings suggest that mice lacking components of the inflammasomes are shielded from HFD-induced weight gain, adipocyte hypertrophy, and hepatic steatosis. To study the role of NLRP3 inflammasomes in NAFLD, both loss-of-function and gain-of-function *Nlrp3* mice models had been produced, and research has confirmed that *Nlrp3* deficiency provides protection against diet-induced steatohepatitis, liver enlargement, liver injury, and activated macrophage infiltration, and vice versa [65]. Still, data from human samples showed increased NLRP3 inflammasome levels in patients with severe NAFLD, in addition to the up-regulated *Il-1β* mRNA and its correlated *Col1a1* mRNA [65]. Furthermore, in *Nlrp3* knock-in mice, the number of caspase-1 and propidium iodide (PI) double-positive cells in hepatocytes increased significantly, resulting in severe liver inflammation and increased collagen deposition in inflammatory areas, showing higher mRNA levels of connective tissue growth factor (*Ctgf*) and tissue inhibitor of matrix metalloproteinase 1 (*Timp1*), two damaged liver markers releasing from hepatic stellate cells (HSCs) [62]. Therefore, it can be inferred that the aberrant activation of NLRP3 inflammasomes induced liver pyroptosis, culminating in severe liver inflammation, HSCs activation, and collagen deposition. These findings suggest that NLRP3 inflammasomes impacts the progression of hepatic steatosis. In addition, NLRP3 inflammasomes also mediate pyroptosis in Kupffer cells (KCs)—macrophages colonizing in the liver that contain and clear harmful exogenous particulates and immunoreactive materials [80]. By stimulating KCs with palmitic acid (PA), it was uncovered that FFAs activated NLRP3 inflammasomes, implying their roles in NASH [66]. Nevertheless, PA, acting as a DAMP, also triggered the release of mitochondrial DNA (mtDNA) from mitochondria to the cytoplasm, and the latter directly bound to NLRP3 inflammasomes, thereby initiating the activation of caspase-1 and the release of the pro-inflammatory IL-1β, which exacerbated NASH [66]. Hence, the occurrence of pyroptosis in KCs provides a missing link between NLRP3 inflammasome activation and NASH. In a separate experiment, the inhibition of NLRP3 was found to attenuate the progression of liver inflammation and fibrosis in mice [11]. Building upon the validation of hepatocyte pyroptosis, Gual et al. [63] uncovered its role in the activation of HSCs and the induction of liver fibrosis; *Nlrp3*^KI^ CreA mice displayed upregulated fibrosis markers, activated HSCs, increased collagen deposition, and elevated α-smooth muscle actin (α-SMA) protein expression in liver after 9-month normal chow. The authors further confirmed that internalized extracellular recombinant NLRP3-YFP inflammasome particles in HSCs brought about increased IL-1β secretion and α-SMA protein expression, and the effect on α-SMA expression in LX2 cells (human HSC line) was eliminated with a pretreatment of internalization inhibitors [63]. It was validated that overactivation of NLRP3 resulting in hepatocyte pyroptosis caused the secretion of inflammasome complexes into the extracellular space, which in turn activated HSCs and produced collagen. Furthermore, cholangiocytes, which play a pivotal role in liver fibrosis, have been demonstrated to generate pro-inflammatory cytokines and compromise the epithelial barrier function upon activation by NLRP3 [67]. NLRP3 overexpression was detected in reactive cholangiocytes in patients and murine models affected by primary sclerosing cholangitis; in vitro, NLRP3 activation stimulated the expression of IL-18 in cholangiocytes, with minimal effects on IL-1β [67]. In *Novosphingobium aromaticivorans*-induced primary biliary cholangitis (PBC), Galectin-3 was found to enhance NLRP3 inflammasome activation, thereby stimulating IL-1β production and worsening the progression of PBC [81]. Subsequently, this might lead to intrahepatic cholestasis, consequently culminating in liver fibrosis. Undoubtedly, NLRP3 inflammasomes play pivotal roles in NAFLD, mediating pyroptosis and influencing the progression from steatosis to NASH and liver fibrosis, as described in Figure 2. In addition to NLRP3, NLRC4 and NLRP6 have been identified to mediate hepatocyte pyroptosis and contribute to the development of NAFLD. The accumulation of free cholesterol in the liver stimulated the expression of sphingomyelin synthase 1 (*Sms1*) in NASH and demonstrated through in vitro experiments that SMS1 exacerbated liver injury, hepatitis, and liver fibrosis by affecting NLRC4-mediated hepatocyte pyroptosis, but had no effect on steatosis [68]. NLRP6, NLRP3, and the effector protein IL-18 could negatively regulate the progression of NAFLD/NASH by modulating the gut microbiota, highlighting the central role of the microbiota in the pathogenesis of heretofore seemingly unrelated NAFLD [82]. There is no denying that the involvement of the inflammasomes, which serves as pivotal molecules in pyroptosis in the pathogenic mechanisms of NAFLD, is unequivocal. In the context of ALD, alcohol is the primary trigger of the disease, activating NLRP3 inflammasomes and causing significant pyroptosis that cannot be ignored. Heo et al. [61] showed that alcohol affected miR-148a regulation in primary hepatocytes and confirmed that forkhead box protein O1(FoxO1) was a transcription factor for miR-148a that inhibited thioredoxin-interacting protein (TXNIP) expression. In vitro experiments using miR-148a transfection inhibited alcohol-induced TXNIP expression, indicating that miR-148a regulates ALD through TXNIP and that alcohol-induced TXNIP overexpression triggered classical pyroptosis and promoted NLRP3 activation [61]. It was concluded that alcohol-induced ALD reduces miR-148a expression in hepatocytes through FoxO1, leading to TXNIP overexpression and NLRP3 inflammasome activation, which induces hepatocyte pyroptosis [61]. Further, inhibiting ATP signaling and uric acid release can reduce the activation of alcohol-induced NLRP3 inflammasomes, improving alcoholic steatohepatitis (ASH) with potential therapeutic implications [69]. In vitro experiments suggested that damaged hepatocytes mainly released ATP and uric acid, which acted as a second signal driving NLRP3 inflammasome release and IL-1β production in ASH, which played a crucial role in LPS-induced liver inflammation of intestinal origin [69]. Collectively, inhibiting alcohol-induced inflammasomes activation may ameliorate ALD, albeit the association with pyroptosis warrants further experimental verification. Nevertheless, among the numerous inflammasomes, NLRP3 assumes a pivotal function in the pathogenesis of ALD and is anticipated to emerge as a promising therapeutic target in the times to come.

### 3.2. Caspases and FLD

Inflammatory caspases mediate the maturation of specific cytokines, forming active IL-1β and IL-18, and their involvement in hepatocyte pyroptosis has been linked to the onset and progression of FLD. Mainly including caspase-1/4/11, the caspases known to participate in hepatocyte pyroptosis, their elimination significantly improves FLD (Figure 2). However, the role of caspase-3/8 in hepatocyte pyroptosis remains unclear. Gaul et al. [63] established the existence of the classical pyroptosis pathway mediated by caspase-1 in hepatocytes. Upon stimulation with LPS combined with PA or nigericin, the number of active caspase-1 and PI-positive cells increased significantly in primary mouse hepatocytes, human hepatocytes, and HepG2 cells; lactate dehydrogenase release also increased in primary mouse hepatocytes and HepG2 cells, which was completely eliminated by blocking caspase-1. The experiment further showed that the NLRP3 inhibitor MCC950 could inhibit caspase-1 activation in HepG2 cells, and GSDMD knockout HepG2 cells and primary mouse hepatocytes showed resistance to the occurrence of pyroptosis [63]. This suggests that in response to NLRP3 activators, hepatocytes undergo classical pyroptosis that is mediated by caspase-1. Immunoblot analysis showed the extracellular release of NLRP3 inflammasome components (NLRP3, ASC, pro-caspase-1, pro-IL-1β), IL-1β, and caspase-1 during hepatocyte pyroptosis [63]. Similar results have also been observed under different dietary inductions. In a NASH mouse model induced by a high-fat and high-cholesterol diet (HFHCD), the absence of caspase-1 provided protection against liver inflammation and fibrosis, while also preventing pyroptosis in hepatocytes [68]. Undoubtedly, this further confirms the involvement of caspase-1 in the progression of NAFLD. Furthermore, the non-classical pathway, which is mediated by caspase-11, was discovered to have an impact on the development of liver steatosis and NASH [70]. In the liver of NASH mice, there was a significant upregulation of both pro-caspase-11 and caspase-11 levels compared to WT mice. Interestingly, *Casp-11* knockout resulted in reduced liver steatosis, swelling, improved liver injury, and inflammation [70]. In mice overexpressing caspase-11, serum levels of ALT, AST, and liver triglycerides increased, along with worsened steatosis and liver injury. In vitro experiments using LPS-induced primary hepatocytes showed that caspase-11 promoted NASH progression by mediating hepatocyte pyroptosis [70]. The study also confirmed that caspase-11 facilitated caspase-1, but unfortunately, the activation of NLRP3 inflammasomes was not detected [70]. However, a study by Gual et al. [68] reported that the deletion of caspase-11 did not lead to a significant improvement in NASH and hepatocyte pyroptosis in a murine model induced by an HFHCD. This discrepancy may be due to the different NASH models induced by distinct dietary regimens. New research suggests that the NAFLD development is determined by gut bacteria [83], while caspase-1/11 has been found to influence the gut microbiome and consequently affect the progression of NAFLD. In the HFD-induced NAFLD model, *Casp-1/11*^−/−^ mice had higher body weight and more severe hepatic steatosis, with a higher diversity of intestinal bacteria but a lower abundance of lactobacilli in the gut microbiota, even under a normal diet [84]. These results indicated that caspase-1/11 exerted a significant influence on regulating the overall hepatic lipid composition as well as the gut microbial community composition, thereby contributed to the pathogenesis of NAFLD. In the pathogenesis of ALD, both the classical pathway molecule caspase-1 and the non-classical pathway molecule caspase-11 exerts a profound impact. Using a Lieber-De-Carli diet with binge feeding, Kai et al. [71] developed a chronic ALD model and found that oroxylin A suppressed liver inflammation and the NLRP3-caspase-1 pyroptosis pathway. In vitro, Zhao et al. established an ethanol-induced hepatocyte pyroptosis model using L02 cells and found that quercetin shielded against ethanol-induced hepatocyte pyroptosis by ameliorating mitochondrial ROS and enhancing PGC-1α-mediated mitochondrial homeostasis in L02 cells [72]. The above findings not only demonstrate the involvement of the classic pyroptosis pathway in ALD pathogenesis but also suggest that the potential interaction between cellular pyrop-tosis and mitochondrial pathways could influence ALD. Alcoholic hepatitis (AH) is considered the progressive stage of ASH, and emerging evidence has linked it with the initiator of non-classical pyroptosis, caspase-4/11. Khanova et al. [74] observed that caspase-4/11 was upregulated in both the AH mouse model and patients, with increased activation of GSDMD, which was not discovered in ASH mice. The absence of caspase-11-inhibited GSDMD activation and hepatocyte death, and a deficiency of anti-IL-18 microbiota exacerbated AH in mice. A comparative study of liver tissues from severely ill AH patients and healthy controls verified the activation of the caspase-4-GSDMD pathway in clinical AH [74]. Therefore, it can be inferred that the non-classical pyroptosis pathway mediated by caspase-4/11 is involved in the pathogenesis of AH, and components of this pathway may serve as new therapeutic targets. It is evident that caspase-1/4/11, by mediating both classical and non-classical cell death pathways, exerts a significant impact on the pathogenic progression of both NAFLD and ALD.

### 3.3. GSDMs and FLD

GSDMs are the ultimate executioners of pyroptosis, which are cleaved by caspases to generate N-terminal fragments with pore-forming activity that determine the mode of cell death. Presently, two GSDMs, namely GSDMD and GSDME, have emerged as key players in the pathogenesis of FLD. GSDMD is closely associated with hepatic steatosis, hepatitis, and fibrosis. Experiments have shown that GSDME can induce pyroptosis and liver inflammation, but its specific role in FLD is still unclear. In methionine-choline deficient diet (MCD)-induced NASH mice, Xu et al. [60] identified that *Gsdmd*^−/−^ mice demonstrated remarkable enhancements in steatosis and liver inflammation. Additionally, the expression of the fat synthesis gene *Srebp-1c* decreased, while the expression of the fat degradation gene *Pparα* and its downstream targets *Aco*, *Lcad*, *Cyp4a10*, and *Cyp4a14* increased, suggesting that GSDMD could exert an influence on hepatic lipid synthesis via the modulation of lipid-related gene expression [60]. The study further demonstrated that the expression of the GSDMD-N terminal fragment was positively correlated with hepatic lobular inflammation and hepatocyte ballooning [60]. This indicates that the GSDMD-N terminal fragment shows promise as a potential biomarker for NASH diagnosis, highlighting the significant role of GSDMD in NASH development by regulating lipogenesis, inflammatory responses, and the NF-κB signaling pathway. The above findings reveal potential therapeutic targets for human NASH. GSDMD is also linked to liver fibrosis [60]. Under MCD feeding conditions, knocking out GSDMD significantly reduced mRNA levels of transforming growth factor-beta1 (*Tgf-β1*) and *α-SMA*, proposing that GSDMD is necessary for the development of hepatic nutritional fibrosis [60]. The caspase-3/GSDME pyroptosis pathway also links to hepatocyte pyroptosis, but its effect on NAFLD has not been experimentally confirmed. After chronic and subacute oral administration of deoxynivalenol (DON), mice liver manifested inflammatory damage, focal steatosis, focal fibrosis and activated caspase-3 as well as GSDME, which were suppressed by the caspase-3 inhibitor Z-DEVD and Ac-DEVD [75]. In vitro, typical pyroptotic characteristics and balloon-like bubbling were observed in HepaRG cells induced by DON, with activation of caspase-3 and GSDME and secretion of IL-1β [75]; knocking down GSDME and inhibiting caspases activity significantly blocked DON-induced pyroptotic characteristics, while over-expressed GSDME played a promoting role [75]. Therefore, it has been established that DON induces caspase-3/GSDME-dependent hepatocyte pyroptosis and its role in DON-induced liver inflammatory injury. In addition to DON, miltirone has been shown to elicit activation of the caspase-3/GSDME pyroptotic pathway in HCC cells, albeit exclusively in vitro. Zhang et al. [85] found that miltirone restrained the cell viability of either HepG2 or Hepa1-6 cells and elicited the proteolytic cleavage of GSDME in each HCC cell. Another experiment proposed a new liver-protective strategy against GSDME-mediated pyroptosis. In vivo and in vitro experiments demonstrated that two new GSDME-derived inhibitors, Ac-DMPD-CMK and Ac-DMLD-CMK, could directly bind to the catalytic domains of caspase-3 and specifically limit caspase-3 activity, and effectively prevent caspase-3/GSDME-dependent hepatocyte pyroptosis and the resulting liver failure [86]. Regrettably, the aforementioned experiments were not conducted in NAFLD models, and the relationship between caspase-3/GSDME-induced pyroptosis and NAFLD remains unknown. Regarding ALD, while GSDMD’s involvement in AH’s pathogenic mechanism has been established, shedding light on the complex transition from ASH to AH, much about other aspects of ALD remains unknown. Khanova et al. [74] reported the upregulation of caspase-11 and GSDMD in an AH mouse model, with no change in caspase-1 expression. Subsequent studies revealed the knockout of the involvement of GSDMD in the caspase-11/4-mediated pyroptosis pathway and its contribution to the development of AH. These findings suggested that GSDMD mediated the pathogenesis of AH by participating in the caspase-11/4-mediated pyroptosis pathway.

In summary, pyroptosis pathways and associated inflammasomes, caspases, and GSDMs play a role in the pathogenesis of FLD. In NAFLD, hepatocytes, Kupffer cells, and hepatic stellate cells are implicated in its occurrence and progression through pyroptosis-induced inflammatory responses. Alcohol-induced liver pyroptosis mediates the pathogenesis of ALD. Exploring the occurrence and impact of different pyroptotic pathways in the liver holds potential for future FLD treatments (Figure 2).

## 4. Strategies for the Prevention and Treatment of FLD with Pyroptosis

Numerous studies have focused on developing therapeutic interventions targeting the pyroptosis pathway, including pyroptosis inhibitors and traditional Chinese medicine, to manage NAFLD/NASH, as summarized in Table 1. Additionally, exercise has emerged as a promising and cost-effective treatment modality for FLD, offering significant potential for future research. One notable finding is the potential candidacy of the MCC950 inhibitor as a drug for NAFLD treatment. Mridha et al. [11] discovered that MCC950 significantly diminished the overexpression of neutrophils and macrophages in NASH mice. In the *foz/foz* mouse NASH model, MCC950 significantly lowered the expression of pyroptosis-related molecules such as NLRP3, caspase-1, pro-IL-1β, and IL-1β, as well as fibrosis-related molecules such as Col1, α-SMA, and CTGF in the liver, bringing about a significant reduction in liver fibrosis [11]. These findings suggested that treatment with MCC950 could improve the inflammatory response in NASH and cure existing liver fibrosis by blocking the expression of liver NLRP3. Additionally, MCC950 can also alleviate cholestasis liver injury. In a mouse model of cholestatic liver injury triggered by bile duct ligation, MCC950 was found to inhibit the infiltration and cell death of neutrophils in the liver, as well as the activation of the NLRP3 inflammasomes induced by cholestasis [87]. This was mainly achieved by inhibiting toll-like receptor signaling pathways to block NLRP3-induced pyroptosis, thus providing some useful insights into the treatment of NAFLD [87]. CY-09, a new NLRP3 inhibitor, was found to lower liver NAFLD activity scores and significantly improve insulin resistance and fasting blood glucose in mice with HFD-induced NASH [76]. IFM-514 is a novel NLRP3 antagonist that has been shown to restrain the development of NASH in *ApoE*^−/−^ mice by suppressing NLRP3 inflammasomes and subsequent caspase-1 protein hydrolytic activation [77]. Traditional Chinese herbal medicines and their related components have also been uncovered to have certain value in the treatment of NAFLD/NASH. Mai et al. [78] found that under the influence of berberine (BBR), lipid accumulation and ROS levels in AML12 cells treated with MCD and LPS in vitro decreased significantly, whereas the mRNA and protein expression levels of TNF-a, as well as the phosphorylation level of NF-κB p65, were significantly lower than those in the PA treatment group. This indicated that BBR can reduce lipid accumulation in hepatocytes, alleviate oxidative stress, and impede the NF-κB signaling pathway and the expression of TNF-a. The experiment also observed that BBR suppressed the protein levels of TXNIP, NLRP3, pro-caspase-1, caspase-1 and GSDMD-N in AML12 cells treated with MCD and LPS/PA, similar to the action of N-acetyl-cysteine [78]. Therefore, it was inferred that BBR suppressed the activation of NLRP3 inflammasomes and pyroptosis through the ROS/TXNIP axis and may be a potential drug for the treatment of NAFLD/NASH [78]. In the MCD-induced NASH model, sweroside treatment significantly reduced IL-1β and caspase-1 expression in macrophages; it also decreased transaminase levels, liver immune cell infiltration, triglyceride accumulation, and fibrosis compared to the control group [79]. These results highlight the potential clinical value of sweroside as a treatment for NASH.

In recent years, exercise has emerged as a low-cost and highly effective therapeutic intervention for NAFLD. Aerobic exercise, resistance training, and physical activity have been shown to mitigate hepatic lipid deposition while ameliorating NAFLD [88,89,90]. Exercise also has a significant anti-inflammatory effect and is closely related to pyroptosis [91]. Studies have indicated that exercise can regulate the transformation of the M1 to the M2 phenotype of adipose tissue macrophages in HFD-induced obese mice, thereby inhibiting inflammation in adipose tissue [92,93]. Aerobic exercise has been evidenced to attenuate the expression of NLRP3 inflammasomes and effectively impede the activation of ASC, caspase-1, IL-1β, and IL-18 [91,94]. In 2020, Yang et al. [8] confirmed that aerobic exercise could suppress NLRP3 inflammasome activation in NASH mice induced by an HFD diet and could also alleviate liver lipid accumulation, inflammation, and fibrosis induced by an HFD or MCD diet in NASH mice. The mechanism may be that exercise increased adropin levels, reduced liver ROS content, and thus suppressed NLRP3 inflammasomes activation. At the same time, another mechanism by which exercise alleviates NAFLD has been discovered. Yu et al. [95] found that aerobic exercise significantly reduced LPS concentration, as well as the mRNA and protein levels of TLR4, MyD88, and NF-κBp65 in HFD-fed mice. Through ApoA5 transfection and LPS intervention studies, they found that exercise significantly reduced serum LPS concentration and increased liver ApoA5 expression, thereby triggering the inhibitory effect on the TLR4-mediated NF-κB pathway [95]. Therefore, exercise may limit NAFLD by suppressing pyroptosis. The role of pyroptosis in ALD pathogenesis has prompted the development of pyroptosis-related therapeutic drugs for ALD. In alcohol-induced AFLD mice, it was discovered that butyrate improved hepatic steatosis, inflammation, intestinal barrier damage, and endotoxemia by downregulating GSDMD-mediated pyroptosis, which may mainly improve alcohol-induced liver injury by maintaining intestinal barrier function and reducing intestinal leakage [96]. In addition, Luan et al. [97] determined the key role of GSDMD pores in the high secretion of IL-1β triggered by chronic ethanol and developed a new treatment plan for ASH using hepatocyte-specific nanobiologics (Glipo-pVAX1-IA) to prolong the expression of IL-1Ra in the liver to prevent the development of ASH. Fu et al. [98] identified that selenium-enriched spirulina played a protective role in alcohol-induced liver injury by reducing caspase-1-induced pyroptosis. Regarding the effect of exercise on alcoholic liver disease, Cui et al. [99] found that an 6-week exercise intervention during the recovery period of ALD effectively improved hepatocyte injury and blood lipid abnormalities, while drinking during exercise exacerbated blood lipid abnormalities and oxidative stress, worsening alcohol-induced liver injury. Therefore, long-term abstinence is the most effective strategy to prevent the progression of ALD, while exercise intervention during the recovery period can effectively alleviate its pathogenic process. The exploration of liver pyroptosis holds promise for the development of clinical and drug therapies for FLD, with the potential emergence of pyroptosis signal pathway inhibitors as novel therapeutic options.

## 5. Summary and Prospects

In summary, classical and non-classical pyroptosis, along with the caspase-3/GSDME pathway, impact FLD pathogenesis. Pyroptosis induces hepatic steatosis in hepatocytes and triggers inflammatory responses in KCs, driving the progression from fatty degeneration to NASH. Furthermore, pyroptosis in hepatocytes and KCs also contributes to HSC activation and liver fibrosis in NAFLD. ALD is characterized by alcohol-induced classical and non-classical pyroptosis, resulting in an inflammatory storm, and this process may be linked to mitochondria. Some strategies targeting pyroptosis, such as NLRP3 inhibitors and exercise, offer potential for FLD treatment. Specifically, resistance and aerobic exercise show promise in mitigating pyroptosis and improving NAFLD outcomes. However, further investigation is needed to understand the impact of pyroptosis in exercise and ALD. The role of GSDME-mediated signaling in the liver remains unclear, but exploring its potential, similar to GSDMD, holds promise for future FLD treatment research.

## Figures and Tables

**Figure 2 ijms-24-13065-f002:**
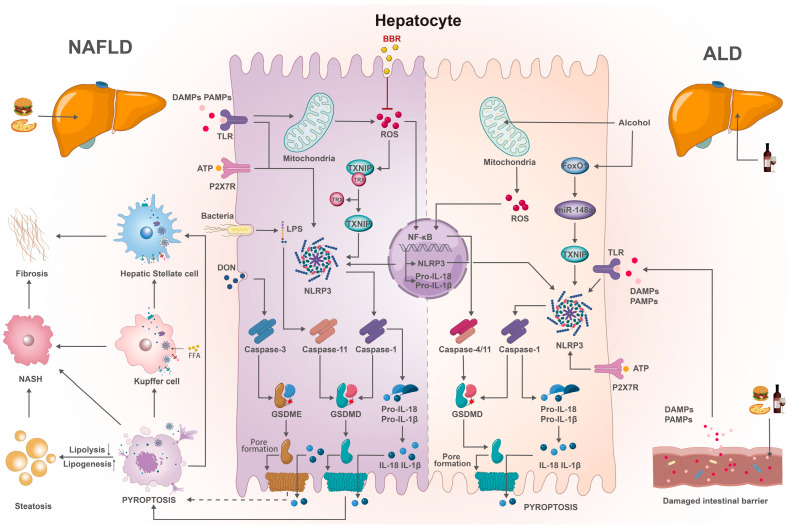
Mechanism of pyroptosis in FLD. In NAFLD, hepatocytes can undergo classical pyroptosis mediated by caspase-1, non-classical pyroptosis mediated by caspase-11, and caspase-3-dependent pyroptosis. Hepatocyte pyroptosis leads to lipid accumulation by inhibiting lipolysis and promoting lipogenesis, thereby causing steatosis. On the other hand, NLRP3 inflammasomes released from hepatocyte pyroptosis are internalized by KCs and HSCs, inducing pyroptosis in each respective cell type. Pyroptosis of hepatocytes and KCs promotes the transition from steatosis to NASH, while pyroptosis of HSC triggers the progression from NASH to liver fibrosis. As for ALD, under the induction of alcohol, hepatocytes undergo caspase-1-mediated classical pyroptosis and caspase-4/11-induced non-classical pyroptosis. In addition, under the influence of high fat and alcohol, DAMPs and PAMPs released from the intestine can also induce hepatocyte pyroptosis in NAFLD and ALD. FLD: Fatty liver disease. NAFLD: non-alcoholic fatty liver disease. ALD: alcoholic liver disease. NASH: non-alcoholic steatohepatitis. DAMPs: damage-associated molecular patterns. PAMPs: pathogen-associated molecular patterns. TLR: Toll-Like Receptors. ATP: adenosine triphosphate. P2X7R: P2X7 receptor. NF-κB: nuclear factor-kappa B. GSDMD/E: gasdermin/D/E. IL: interleukin. LPS: lipopolysaccharides. NLRP3: NLR family pyrin domain-containing 3. DON: deoxynivalenol. TXNIP: thioredoxin-interacting protein. ROS: reactive oxygen species. TRX: Thioredoxin. FoxO1: forkhead box protein O1. miR: MicroRNA. FFA: free fatty acid. BBR: berberine.

**Table 1 ijms-24-13065-t001:** Researches on pyroptosis and FLD.

Pyroptotic Component	FLD Feature	Experimental Model	Results (Compared with Controls)	Reference
NLRP3	Steatosis	HFD-fed *Nlrp3*^−/−^, *Asc*^−/−^ and *Casp1*^−/−^ mice	↓ Steatosis, ↓ weight gain, and ↓ hepatic triglyceride content	[64]
	Steatosis	CDAA-fed *Nlrp3*^−/−^ mice	↓ Steatohepatitis, ↓ liver enlargement, ↓ liver injury, ↓ liver fibrosis, and ↓ activated macrophage infiltration	[65]
	Steatosis	*Nlrp3* knock-in mice	Severe liver inflammation, hepatic stellate cell activation and collagen deposition	[62]
	NASH	MCD-fed *Nlrp3*^−/−^ mice	↓ Steatosis, ↓ hepatocyte ballooning, ↓ inflammatory cell infiltration, ↓ expression of NLRP3, ASC and caspase-1 in KCs	[66]
	NASH	PA stimulates primary KCs from MCD-fed *Nlrp3*^−/−^ and WT mice	NLRP3 inflammasome complex formation, the colocalization of NLRP3 with caspase-1, caspase-1 activation and IL-1β secretion	[66]
	NASH	MCC950-treated MCD-fed mice and atherogenic diet-fed *foz/foz* mice	Normal hepatic caspase 1 and IL-1β expression, ↓ ALT/AST, ↓ the severity of NASH pathology and liver fibrosis	[11]
	NASH	LPS plus Nig or LPS plus PA primary mouse and human hepatocytes	↑ Caspase-1- and PI-positive cells, NLRP3 inflammasome proteins release	[63]
	NASH	Hepatocyte-specific leucine 351 to proline Nlrp3^KI^CreA mice	↓ Upregulated fibrosis markers, ↓ collagen deposion and ↓ α-SMA protein expression	[63]
	Liver fibrosis	LPS and ATP stimulates cholangiocyte	↑ IL-18 expression,↓ E-cadherin and Zonulin-1 protein expression	[67]
NLRC4	NASH	HFHCD-fed mice	↑ The expression of SMS1, NLRC4 and simple steatosis without significant inflammation and fibrosis	[68]
IL-1β	ASH	Lieber-DeCarli ethanol-fed P2rx7-KO and overexpress uricase mice	ASH attenuation, ↓ inflammasome activation and ↓ IL-1β level	[69]
Caspase-1	Liver fibrosis	HFHCD-fed *Casp1*^−/−^ mice	↓ Hepatic inflammation and ↓ fibrosis	[68]
Caspase-11	NASH	MCD-fed *Casp11*^−/−^ mice	↓ Liver injury, ↓ fibrosis, ↓ inflammation, ↓ GSDMD and IL-1β activation	[70]
NLRP3, Caspase-1	ALD	Lieber-De-Carli diet with binge-fed ALD mice	↑ Inflammatory infiltration, ↑ intracellular TG and TC, ↑ caspase-1, IL-18, IL-1β, NLRP3 activation	[71]
	ALD	Ethano and quercetin-treated lL02 cells	↓ MtDNA production, ↓ NLRP3, ASC, caspase1, IL-18, IL-1β and GSDMD-N expression	[72]
Caspase-11/4	AH	Hybrid-fed AH mice [73]	↑ Caspase-11/4, ↑ GSDMD and ↓ IL-1β levels	[74]
Caspase-11	AH	Hybrid-fed AH *Casp11*^−/−^ mice	↓ GSDMD level, ↓ hepatocellular death, ↓ liver bacterial load	[74]
IL-18	AH	Hybrid-fed AH *IL-18*^−/−^ mice	↑ GSDMD activation, ↑ liver bacterial load and ↑ hepatocyte death	[74]
GSDMD	NAFLD	Human liver tissues from NAFLD patients	↑ GSDMD and GSDMD-N fragment level	[60]
	NASH	MCD-fed *GSDMD*^−/−^ mice	↓ Steatosis, ↓ inflammation, ↑ lipogenic gene expression and↑ lipolytic genes expression	[60]
GSDME	Liver inflammatory injury	DON orally administered mice	Focal steatosis, focal fibrosis and caspase-3, PARP and GSDME activation	[75]
	Liver inflammatory injury	DON-exposed HepaRG cells	↓ Cell viability and ↑ Annexin-V/PI double positive cells	[75]
NLRP3	NAFLD	CY-09-treated HFD-fed mice	↓ Body weight, ↓ liver lipid droplets and ↓ NAS	[76]
	NASH	IFM-514 -injected MCD-fed *ApoE*^−/−^ mice	↓ IL-1β production, ↓ caspase-1 activation, ↓ NAS and ↓ steatosis	[77]
	NASH	BBR-treated MCD Medium-induced AML 12 cells	↓ Lipid accumulation, ↓ ROS, ↓ lipid peroxides, ↓ NLRP3, caspase-1 and GSDMD-N expression	[78]
	NASH	Sweroside-treated MCD-fed mice	↓ IL-1β, ↓ caspase-1, ↓ hepatic immune cell infiltration, ↓ hepatic triglyceride accumulation, and ↓ liver fibrosis	[79]
	NASH	Exercise-intervened HFD-fed or MCD-fed mice	↓ NLRP3 inflammasome components, ↓ caspase-1 enzymatic activity, ↓ hepatic steatosis, ↓ inflammation, and ↓ fibrosis	[80]

HFD: high-fat diet; CDAA: choline-deficient amino acid-defined; MCD: methionine-choline deficient diet; NASH: nonalcoholic steatohepatitis; ASH, alcoholic steatohepatitis; ALD: alcoholic liver disease; NAFLD: non-alcoholic FLD; AH: Alcoholic hepatitis; KCs: Kupffer cells; WT: wild type; ALT: alanine aminotransferase; AST: aspartate aminotransferase; LPS: lipopolysaccharide; Nig: nigericin; PA: palmitate; PI: propidium iodide; α-SMA: a-smooth muscle actin; ATP: adenosine triphosphate; HFHCD, high-fat and high-cholesterol diet; SMS1: sphingomyelin synthase 1; TG: triglyceride; TC: total cholesterol; DON: deoxynivalenol; PARP: poly (ADP-ribose) polymerase; NAS: NAFLD activity score; BBR: berberine; ROS: reactive oxygen species.

## Data Availability

Not applicable.

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
