# Peer review of "Research Progress of Pyroptosis in Fatty Liver Disease"

_ijms, 2023, doi:10.3390/ijms241713065_

Round 1

Reviewer 1 Report

Li et al's review focused pyroptosis in FLD. The authors logically explained the components associated with pyroptosis and mentioned the relationship between pyroptosis and FLD from basic research to medicinal approach. 

As major comments, the review has only two figures otherwise there many molecules/components are mentioned. I want the authors to indicate some tables, including the differences between pyroptosis and apoptosis, caspases associated with pyroptosis, GSDMDs, inflammasomes components and so on. And please put more figures, including image diagram of inflammasomes. 

In Figure1, can the authors split into 2 images, including the canonical and non-canonical pathways? Is the arrow from hypoxia to caspase-8 correct in Ref54?

Minor comments

Line 3 of Abstract, "its derivatives is ..." is " its derivatives are ...".

Page2 Line48, "interleukin (IL)-1β" might be corrected to IL-1β.

There are nouns in the middle of a sentence that start with a captal letter, occasionally.  For example, Page3 Line 134, Page4 Line 159, Line 166.

Page4 Line 167 "K+" is "K+". 

Page9 Line 415, "Ac-DMLD- CMK" has a wasted space between - and C.

 In Figure2, there is BBR on the top of the figure. In text, BBR is explained berberine in Page11. The explanation might also be put in legend of Figure2.

In Reference, please check how to indicate journal title. The manuscript show its all references with capital letters.  

The English of manuscript is easy to read and understand the contents. 

Reviewer 2 Report

In this review, Li et al. summarize the roles of pyroptosis in fatty liver disease. This review is well written, covering the important fields of this hot topic, showing good figures. I have only minor comments for this manuscript.

·       Multiple errors are found in this study, especially abbreviation. Careful proofreading should be performed entirely.

·       If the authors used the third-party service to draw figures, such as Biorender, information for that service should be shown (name, location, or website address).

·       In Figure 1, GAS looks like a cell, but this is bacteria. In Figure 2, DON looks like bacteria, but this is not. “Alcoholic” should be “Alcohol”, and “Teatosis” should be “Steatosis”.

·       Hepatocytes are the major player for the pathogenesis of ALD and NAFLD, but recent studies showed that other hepatic cells, such as cholangiocytes and hepatic stellate cells, are also important. This manuscript covers a little about stellate cells, but not information for cholangiocytes. Why not pyroptosis in cholangiocytes and stellate cells? During liver damage, especially in cholestatic liver injury, cholangiocytes and stellate cells secrete cytokines, such as IL-6 and TGFb1, leading to liver fibrosis. Functional roles of these cells in fatty liver disease are not fully understood, these cells may be involved in pathogenesis of FLD or pyroptosis. It would be more useful for readers if the authors discuss about the roles of other hepatic cells.

minor
